# Modification Role of Dietary Antioxidants in the Association of High Red Meat Intake and Lung Cancer Risk: Evidence from a Cancer Screening Trial

**DOI:** 10.3390/antiox13070799

**Published:** 2024-06-30

**Authors:** Jiaqi Yang, Xiaona Na, Zhihui Li, Ai Zhao

**Affiliations:** 1Vanke School of Public Health, Tsinghua University, Beijing 100084, China; jyang211@jhmi.edu (J.Y.); nxn21@mails.tsinghua.edu.cn (X.N.); zhihuili@mail.tsinghua.edu.cn (Z.L.); 2Department of Epidemiology, Johns Hopkins Bloomberg School of Public Health, Baltimore, MD 21205, USA; 3Institute for Healthy China, Tsinghua University, Beijing 100084, China

**Keywords:** red meat, lung cancer risk, dietary antioxidants, supplements, effect modification, reduced rank regression

## Abstract

Evidence on the association between red meat consumption and lung cancer risk is weak. This study examined the associations between red meat and lung cancer across levels of antioxidant intake from foods or supplements. Cox proportional hazard models were applied to assess hazard ratios (HRs) for lung cancer incidence in the Prostate, Lung, Colorectal, and Ovarian (PLCO) cancer screening trial. Baseline food frequency questionnaires measured red meat and antioxidant intake. The food-based Composite Dietary Antioxidant Index (fCDAI) evaluated the overall natural intake of vitamin A, vitamin C, vitamin E, zinc, magnesium, and selenium. During 13 years of follow-up, 95,647 participants developed 1599 lung cancer cases. Higher red meat consumption was associated with a higher risk of lung cancer (HR_Q4vsQ1_ 1.43, 95%CI 1.20–1.71, *p*-trend < 0.001). We observed similar trends across groups with low or medium levels of antioxidant intake. However, no association was noticed in the group with the highest fCDAI (HR_Q4vsQ1_ 1.24, 95%CI 0.90–1.72, *p*-trend = 0.08) and highest independent natural antioxidant intake. The attenuated risk was not consistently observed among groups with high supplement use. Lastly, we did not notice evidence of interactions between red meat and antioxidant intake. Our findings emphasize the importance of limiting red meat in lung cancer prevention.

## 1. Introduction

Based on 2023 statistics from the American Cancer Society, lung cancer is the second most common cancer and the leading cause of cancer death in the United States [1]. The risk factors of lung cancer have been well studied. Etiologically, lung cancer is considered a reactive oxygen species-dependent lung disorder [2]. While smoking and indoor air pollution are the established causes of lung cancer development, other factors, such as genetic, environmental, and lifestyle factors, are all associated with lung cancer risk [3,4,5]. In particular, the role of diet in lung cancer remains inconclusive, especially for food items with pro-oxidant and antioxidant properties.

Red meat provides essential nutrients to the human body. In contrast, red meat consumption is prone to lipid oxidation due to its high content of saturated fat and heme iron. During meat processing and preservation, high-temperature cooking can introduce mutagenic byproducts, such as N-nitroso compounds, which could destabilize the normal oxidation balance in the body by inducing oxidative stress, damage cell structures, and contribute to lung carcinogenesis [6,7,8]. The World Cancer Research Fund (WCRF) and the American Institute for Cancer Research suggest that there is strong evidence for limiting red meat consumption to less than three portions per week (350–500 g for cooked weight) for colorectal cancer prevention purposes [9]. For lung cancer, the WCRF conveys some evidence showing that red meat is closely related to lung cancer risk increment. Still, the evidence is inconsistent as individuals respond differently to red meat and oxidation [10,11,12].

In contrast, foods high in antioxidant vitamins and flavonoids have been associated with a lower risk of lung cancer [13]. We previously reported that a higher food-based Composite Dietary Antioxidant Index (fCDAI), including vitamin A, vitamin C, vitamin E, zinc, selenium, and magnesium, was associated with lower lung cancer risk in the US population [14]. Retinoids, the derivatives of vitamin A, are hypothesized to modulate reactive oxygen species, with antiproliferative effects. Vitamin C also holds free radicals and protects cells from oxidative DNA damage. Considering the capacity of antioxidants, synthetic and natural antioxidants are commonly used in the food industry to preserve red meat from oxidation [15]. However, for dietary intake, the evidence for the modification effect of antioxidants on red meat and lung cancer association is still lacking [16]. One Swedish cohort study found no interaction between different levels of red meat and fruit and vegetable intake on cardiovascular disease and all-cause cancer mortality, with 16 years of follow-up [17]. In a Korean cancer screening cohort with 8024 subjects, researchers reported that dietary flavonoid intake was beneficial in cancer prevention in the low red meat consumption group (<43 g/day) but not in the high red meat consumption group (>43 g/day) [17]. These results suggested that dose-response modification effects of antioxidants might exist in red meat consumption. Yet, varied antioxidants, such as vitamins and minerals, were not individually examined. Moreover, previous studies mainly focused on antioxidants obtained from foods and rarely compared them with supplemental antioxidants, which may have different properties and food synergic effects on lung cancer development [14,18,19]. To the best of our knowledge, no study has studied the potential modification effects of antioxidants from different sources on associations between red meat and lung cancer risk.

It should be noticed that, in 2022, a US cross-sectional study used 2015–2016 and 2017–2018 waves of National Health and Nutrition Examination Survey data and reported that the mean intake of red and processed meat in the US was 105 g/day, which exceeded the world’s average consumption (68 g/day) and World Cancer Research Fund’s recommendation (350–500 g/week) [20]. Given the high red meat consumption and lung cancer rates in the US, further research is needed to elucidate whether dietary antioxidant intake can modify the association between red meat and lung cancer risk.

We, therefore, conducted the present study to prospectively investigate the associations between red meat consumption and lung cancer risk across levels of antioxidant intake from food and supplements, using data from the Prostate, Lung, Colorectal, and Ovarian (PLCO) cancer screening trial. In addition, we devised an antioxidant-rich dietary pattern to further validate our findings.

## 2. Materials and Methods

### 2.1. Study Population

We obtained data from the PLCO cohort, a large randomized controlled trial in the US, to investigate the effects of screening exams on prostate, lung, colorectal, and ovarian cancer incidences and mortalities. Between 1993 and 2001, approximately 155,000 men and women, aged 55 to 74, free of cancer at baseline, were enrolled in the trial across 10 US study centers. After enrollment, all participants were asked to complete a baseline questionnaire for sociodemographic information collection. The dietary history questionnaire (DHQ) was introduced in 1998 and administered to the controlled arm (usual care) and intervention arm (cancer screening).

We extracted 154,887 participants from the lung dataset in the PLCO trial for our analyses. We excluded participants who failed to complete an eligible and valid baseline questionnaire and DHQ (*n* = 53,155). An eligible questionnaire was completed by participants who had no cancer history before the trial or the questionnaire and had time to develop their first cancer incidence following questionnaire completion. A valid questionnaire for dietary analysis was identified by an available date of completion, a date of completion before the date of death, less than 8 missing frequency responses, and no extreme calorie intake (top 1% and bottom 1%). We also excluded participants with missing or unknown covariates, including age, sex, race, study arm, body mass index, education, marital status, total energy intake, pack-years of smoking, alcohol drinks per day, and family history of any cancer (*n* = 6085). In total, 95,647 participants were included in the final analysis (Appendix A).

This Cancer Data Access System project was approved by the National Cancer Institute (NCI), and the project ID is PLCO-974. Written consent forms were obtained from all participants in the PLCO screening trial.

### 2.2. Assessment of Red Meat and Dietary Antioxidant Intake

Total red meat intake was assessed by the DHQ by summing up gram amounts of red meat. For mixed dishes, grams counted for red meat were calculated based on the red meat percentage of each food item.

We obtained dietary and supplemental antioxidant intake from the DHQ for six major antioxidants, including vitamin A (IU/day), vitamin C (mg/day), vitamin E (mg/day of alpha-tocopherol equivalents), magnesium (mg/day), selenium (mcg/day), and zinc (mg/day). Nutrients were calculated using DietCalc software (1.4.3) and linked to a nutrient database based on Nutrition Data Systems for Research or the US Department of Agriculture Survey Nutrient Database. In our study, fCDAI, a summary score for dietary intake of six antioxidants, was also calculated [14]. It has been negatively associated with pro-inflammatory biomarkers in previous studies [21,22]. The calculation of fCDAI for each participant used the following formula:$$\mathrm{fCDAI}=\sum_{i=1}^{6}\frac{xi-\mu i}{Si}$$
where *xi* refers to the daily consumption of antioxidants *i*. *μi* refers to the average consumption of antioxidant *i* in the study population, and *Si* refers to the standard deviation from *μi* of antioxidant *i*.

### 2.3. Covariates Ascertainment

Participants self-reported on demographic characteristics, including age, sex (male or female), race (non-Hispanic White, non-Hispanic Black, Hispanic, Asian, Pacific Islander, or American Indian), study arm (screening or usual care), and levels of education (<8 years, 8–11 years, 12 years or completed high school, post-high school training other than college, some college, college graduate, or postgraduate or more). Other health behaviors self-reported in baseline questionnaires included marital status (married, widowed, divorced, separated, or never married), body mass index (weight in kilograms divided by height in meters squared) generated from height and weight, smoking history (pack-years of cigarettes and smoking status by never, current, or former users), alcohol use (never, current, or former users), and family history of any cancer (yes or no). Total energy intake and supplement use were assessed by the DHQ. We generated the use of any supplement as an additional covariate by examining the self-reported supplement use of vitamin A, vitamin C, vitamin E, zinc, selenium, and magnesium (yes or no).

### 2.4. Case Ascertainment of Lung Cancer and Follow-Up

The primary outcome of our study was lung cancer incidence. Participants in the intervention group underwent chest X-ray screening at baseline, and each participant was offered at least 2 screens. Participants with a positive chest radiographic result with suspected abnormalities were asked to seek a diagnostic evaluation. During 13 years of follow-up, cases were ascertained by contacting suspected participants for medical record abstraction, self-reported questionnaires, and linkage to the National Death Index [23].

### 2.5. Statistical Analysis

Primary analyses examined the association between quartiles of red meat intake and lung cancer incidence across levels of antioxidant intake. Antioxidant intake was categorized by (1) fCDAI, calculated from the natural intake of six antioxidants (vitamin A, vitamin C, vitamin E, zinc, selenium, and magnesium), (2) the independent intake of six antioxidants from foods, (3) the independent intake of six antioxidants from supplements, and (4) the independent intake of six antioxidants from foods and supplements. We used Cox proportional hazard regression in all models to evaluate the associations between quartiles of red meat intake and the risk of lung cancer. To test the linear trend, we used a term for quartile number and included it as a continuous variable in the models. We examined the assumption of proportionality of the hazards using Schoenfeld residuals. In regression models, we adjusted baseline covariates, including age, total energy intake, pack-years of smoking, and alcohol drinks per day as continuous variables; sex (male or female), race (White, Black, Hispanic, Asian, Pacific Islander, or American Indian), study arm (controlled arm or intervention arm), body mass index category (underweight, normal, overweight, or obese), education level (less than high school, high school or post-high school training, or some college or more), marital status (married; widowed, divorced, or separated; or never married), and family history of any cancer (yes or no) were analyzed as categorical variables. Red meat consumption was examined by quartile in all models, and the lowest quartile (<27 g/day) of red meat intake was the reference group. In models examined across levels of fCDAI or independent antioxidant intake from food, we additionally adjusted for the use of any supplements (yes or no). Similarly, in models examined across levels of supplements, we additionally adjusted for fCDAI. The statistical interaction between red meat and antioxidant intake was assessed by the likelihood ratio test (LRT), where a product term was included. The significance threshold was adjusted through the Bonferroni method.

For secondary analyses, we performed subgroup analyses by sex (male or female), smoking status (never, current, or former), and family history of any cancer (yes or no), using Cox proportional hazard regression, to examine the HRs of red meat intake across levels of fCDAI.

Lastly, reduced rank regression (RRR) was carried out to identify the dietary pattern that explained the maximum variation in the fCDAI from daily food sources. Predictor variables were 22 food groups selected based on culinary usage and nutrient content of daily food consumption. Response variables were vitamin A, vitamin C, vitamin E, zinc, selenium, and magnesium intake from food. Then, we examined HRs of lung cancer to red meat intake, stratified by levels of the first dietary pattern score derived by RRR to assess effect modification, and applied LRT to find statistical evidence of interaction between red meat and the diet score.

All statistical analyses were performed by R Studio (4.3.0, Boston, MA, USA) and SAS Studio on SAS OnDemand for Academics (3.81, Cary, NC, USA). All tests were two-sided, and *p* < 0.05 was statistically significant.

## 3. Results

Baseline characteristics of the study population are displayed in Table 1, categorized by quartile of red meat consumption. The mean age of the total sample was 62; 31% was female, and 91% was non-Hispanic White. The group with the highest intake of red meat was predominantly male, non-Hispanic White, more likely to be overweight or obese, less educated, and married. They also had a high amount of tobacco smoking exposure over time. Compared to the lower red meat intake groups, the high-intake group had a higher alcohol intake, fCDAI, and total energy intake and was less likely to take supplements.

There were 1599 lung cancer cases developed during 1,142,192 person-years of follow-up. A higher intake of red meat was associated with a higher risk of lung cancer after controlling for baseline covariates (*p*-trend < 0.001) (Table 2). After stratification by levels of fCDAI (Figure 1, Appendix A) and each antioxidant intake from foods and supplements (Figure 2, Appendix A), we observed similar linear dose-response associations between red meat and lung cancer risk. Yet, red meat consumption was not associated with an increased risk among individuals with the highest fCDAI (HR_Q4vsQ1_ 1.24, 95%CI 0.90–1.72, *p*-trend = 0.08). Similarly, we did not observe an increased risk among individuals with the highest intake of any of the six antioxidants from foods.

In contrast, for people taking a high level of antioxidant supplements, the associations were inconsistent across supplements. Red meat intake was not associated with increased lung cancer risk among the group with high fat-soluble supplement use of vitamin A (HR_Q4vsQ1_ 1.35, 95%CI 0.99–1.84, *p*-trend = 0.037) and vitamin E (HR_Q4vsQ1_ 1.26, 95%CI 0.92–1.72, *p*-trend = 0.219) (Figure 2a, Appendix A). Yet, we observed significant associations of red meat intake with lung cancer risk in groups with high water-soluble (vitamin C) and mineral (zinc, magnesium, and selenium) supplement use (Figure 2b,c, Appendix A).

When stratified by levels of total antioxidant intake, from both foods and supplements, we did not find associations between high red meat intake and elevated lung cancer risk among groups with the highest fat-soluble vitamin intake (vitamin A and vitamin E) but did find associations for individuals with low and medium fat-soluble vitamin intake (Figure 2a, Appendix A). For vitamin C and mineral antioxidants, we found inconsistent linear dose-response associations across different levels of antioxidant intake.

No evidence of statistical interaction was found between red meat and fCDAI (Table 3), nor among each antioxidant intake (from foods, supplements, or both) after the Bonferroni correction (Table 4). All HRs of the interaction term were less than 1 (Table 3).

For secondary analyses, we evaluated the associations between red meat intake and lung cancer risk across levels of fCDAI in each subgroup (Figure 3). The lowest quartile of the red meat consumption group was the reference. Consistent with previous findings in the total sample, higher red meat intake was associated with a higher lung cancer risk when the fCDAI was low and medium, and the association was most pronounced among males, former smokers, and individuals with a family history of cancers (Figure 3a,e,f). Yet, when the fCDAI was high, the association disappeared among those subgroups.

We devised an antioxidant-rich diet score. The RRR analysis identified a major dietary pattern that explained 51% of the variation in response variables. The Spearman correlation between six response variables was depicted (Appendix A), and the factor loadings of the dietary pattern were displayed (Appendix A). The dietary pattern was typified by a diet highest in total vegetable intake, followed by non-starchy vegetables, total meat, red meat, total fruits, white meat, and total fish intake. Similar to previous findings, we did not observe evidence of statistical interaction between red meat intake and diet score in the total population (*p*-interaction = 0.557) (Appendix A). In the stratification analysis by levels of diet score, linear dose-response associations between red meat and lung cancer risk were observed across all levels of diet scores (*p*-trend < 0.05). Yet, similar to the previous findings, in the group with the highest diet scores, no association was shown between red meat consumption and lung cancer risk (HR_Q4vsQ1_ 1.34, 95%CI 0.97–1.85, *p*-trend = 0.026).

## 4. Discussion

In this prospective study among older adults in the US from the PLCO cancer screening trial, we investigated the associations between red meat consumption and lung cancer risk across levels of antioxidant intake, including fCDAI, vitamin A, vitamin C, vitamin E, zinc, selenium, and magnesium, from food and supplements, and further examined the interaction between red meat and antioxidant intake. Red meat was associated with an increased risk of lung cancer, and the linear dose-response association was shown across levels of antioxidant intake. We did not find evidence of a statistical interaction between red meat and antioxidant intake. Yet, no association between red meat and lung cancer was noticed in the group with a high level of natural antioxidant intake, indicating an attenuation of lung cancer risk. Lastly, we identified a dietary pattern high in fruits and vegetables in reflection of fCDAI and further validated our findings.

The association between high red meat consumption and lung cancer risk has been reported previously [16,24,25,26]. However, there is limited evidence of effect modification by antioxidant intake. A past meta-analysis identified a dose-response relationship: every increment of 120 g of red meat intake per day was associated with a 35% increment in lung cancer risk [16]. Past studies have also revealed that dietary patterns high in fruits and vegetables and low in red meat have been linked to a reduced risk of lung cancer [24]. Yet, the interaction between red meat and fruit or vegetable intake in predicting cancer incidence or mortality has not been observed [17]. Fruits and vegetables are high in antioxidant nutrients, and antioxidants have been found to be beneficial in preventing lung cancer risk [14]. Similar to previous findings, our study reported associations between higher red meat consumption and higher risk of lung cancer across levels of antioxidant intake, with no interaction between red meat and antioxidant intake identified.

Oxidation is part of normal body metabolism, while the excess intake of red meat can induce oxidative stress, a key contributor to inflammatory diseases, which can eventually develop all types of cancers [27]. Theoretically, antioxidants have the ability to counteract oxidation products by suppressing the formation of free radicals, scavenging active radicals, and removing oxidatively modified proteins [28]. However, we did not observe the expected evidence of interaction between red meat and antioxidants. This could be due to the fact that there were not enough antioxidants to quench the free radicals. The stratified analyses by levels of fCDAI indicated that the associations with increased risk of lung cancer were shown in low- or medium-fCDAI groups. Yet, when the fCDAI was high, no associations were observed, regardless of the amount of red meat consumption. Therefore, our findings indicated the potential of high antioxidant intake from food to modulate the oxidation-induced lung cancer risk.

Natural antioxidants from foods have a favorable impact on attenuating oxidative damage and preventing disease development. Previous evidence has elucidated the complex synergy between foods, which affects the pro-oxidant or antioxidant outcomes [29]. During the cooking process, antioxidants’ activities decrease, and red meat’s lipid oxidation increases under high heating temperatures [30]. Therefore, less oxidation is produced from less processed red meat. During digestion, high concentrations of hydrophilic antioxidants, like dietary vitamin C, decrease lipid oxidation from red meat by balancing reactive oxygen species in the water compartment [29]. Similarly, high concentrations of lipophilic antioxidants, like dietary vitamin A and vitamin E, decrease lipid oxidation by modulating reactive oxygen species in the fat compartment and protect cell membranes from oxidative damage [29]. Our study observations mirrored the mechanisms. When the intake of natural antioxidants was high, no association was identified between red meat consumption and lung cancer risk. In addition to the effects of individual antioxidants, interactions between different antioxidants may produce synergic effects on attenuating lung cancer risk [14]. Thus, our findings suggested that high antioxidant intake from food could be a strategy to balance the oxidation from red meat and lower the risk of lung cancer. However, an adequate amount of antioxidants is necessary to neutralize the free radicals generated by red meat-induced oxidative stress.

In addition to the modulation of natural antioxidants, we did not observe similar favorable effects from antioxidant supplements, especially mineral supplements. Minerals have shown inconsistent relationships with cancer risk, and some micronutrients have exhibited U-shaped associations where an imbalanced diet is noted [31,32]. Red meat is a good source of zinc, and a high-red-meat diet with additional zinc supplements was linked to elevated lung cancer risk, given the high level of overall zinc intake. Differences in zinc uptake from foods and supplements may also impact lung cancer risk, as zinc supplements are prone to excessive usage among adults [32,33]. In addition, zinc functions more as a cofactor for antioxidant enzymes rather than directly neutralizing free radicals, compared to the direct scavenging action of vitamins A, C, and E [34]. In general, though we did not demonstrate the effect modification from antioxidant intake, our results showed that the red meat associations with lung cancer differed by sources of antioxidant intake. Specifically, high levels of natural antioxidants help neutralize oxidation in the body, while supplemental antioxidants are not protective, especially for mineral supplements like zinc.

Our findings were further validated by RRR analysis, where a dietary pattern in reflection of fCDAI was identified. Consistent with previous findings on high levels of natural antioxidant intake, no association was found between red meat and increased lung cancer risk in people following a highly antioxidant-rich diet. Meanwhile, no evidence of statistical interaction was found. It is also worth mentioning that, according to factor loadings of daily food sources, total vegetable intake, non-starchy vegetable intake, and total meat intake were the largest three contributors to explaining the maximum variation of fCDAI. The results suggested that, in addition to vegetables, meat was also a good source of dietary antioxidants. Since we could not obtain granular data on food sources of antioxidant intake, it would be difficult to differentiate the effects of plant-derived and animal-derived antioxidants.

It is important to note that the response to red meat intake depends on who consumes it. Tobacco smoking is an established risk factor for lung cancer development in men and women, which could play a significant role in red meat–lung cancer associations [1,35]. In our study, we found that individuals with the highest red meat intake had a high amount of tobacco smoking exposure over time. Since tobacco smoke contains harmful chemicals like polycyclic aromatic hydrocarbons, which can damage cellular targets and induce oxidative stress, smoking is causally linked to lung cancer development [36]. To isolate the effects of smoking-induced oxidation, we adjusted pack-years of smoking as a confounding factor in all models and conducted subgroup analyses by smoking status. In the subgroup of former smokers, we noticed an attenuation of lung cancer risk when the fCDAI was high, which was not seen in the current or never smokers. In a previous National Health and Nutrition Examination Survey (NHANES) cross-sectional study, researchers found an association between high unprocessed red meat intake and serum C-reactive protein in past smokers, but this was not observed in current or never smokers [12]. Since quitting smoking will not recover the disturbed iron homeostasis system and chronic inflammation caused by smoking in the past, our results added evidence that individuals with chronic inflammation at baseline would benefit more from dietary antioxidant consumption in preventing lung cancer [12]. In addition to former smokers, the benefit from high dietary antioxidant intake was also observed in several subpopulations, including males and individuals with a family history of cancers, when red meat consumption was high. Cultural norms and gender roles shape dietary preferences, with men consuming more red meat compared to women, so the benefit of high antioxidant intake was more pronounced in the male group. In individuals with a family history of cancers, the attenuated lung cancer risk observed in the high dietary antioxidant intake group indicated the potential roles of dietary antioxidants in regulating specific genes and pathways in lung cancer prevention [37]. Future studies are essential to elucidate the underlying mechanisms between lifestyle behaviors, genetic factors, and antioxidant intake in lung cancer prevention and treatment to support individualized dietary interventions [38,39].

### Strengths and Limitations

Of note, our study has several advantages. First, the PLCO trial had a prospective design that could establish temporality in study findings. Second, the large sample cohort and comprehensive subgroup analyses made the results generalizable to the older population in the US. Third, the long follow-up period allows the outcome to have enough time to develop and to be ascertained.

Several limitations should be acknowledged as well. One major limitation of the study was that dietary intake information was obtained from the DHQ. The exposure was self-reported data, which was vulnerable to recall bias and increased the risk of misclassification, especially for dietary supplement usage [40]. Since the accuracy of case ascertainment was not different across exposure status, and the DHQ has been proven valid previously, the misclassification of non-differential measurement errors would only attenuate the estimates of association we have found [41]. Moreover, attention should be paid when interpreting findings on selenium. The amount of selenium in food depends significantly on the selenium content of the soil where the food is grown, making it challenging to estimate intake accurately solely from dietary questionnaires [42,43]. In addition, because the questionnaires were only collected once at the baseline, we were unable to track changes in diets throughout follow-up. However, dietary changes could be caused by poor health conditions, which would potentially induce reverse causation bias. Lastly, our study did not differentiate the sources of antioxidants (plant-derived or animal-derived) or incorporate phytochemicals with antioxidant properties in the analysis, which may show strengthened modification effects of plant-derived compounds on red meat in lung cancer prevention. However, we were unable to retrieve such granular dietary data from the database.

## 5. Conclusions

In conclusion, our study found that higher red meat consumption was associated with elevated lung cancer risk, and supplemental antioxidant intake could not serve as a remedy for high red meat consumption. Though individual responses to red meat consumption vary, following an antioxidant-rich diet is recommended for lung cancer prevention by attenuating red meat-induced oxidation. Yet, the benefit of natural antioxidants depends on the types of antioxidants and individual characteristics. Therefore, lowering red meat intake should be prioritized as a modifiable risk factor for lung cancer prevention. Future research should explore the roles and interactions of dietary antioxidants and pro-oxidants to provide insights into lung cancer prevention mechanisms.

## Figures and Tables

**Figure 1 antioxidants-13-00799-f001:**
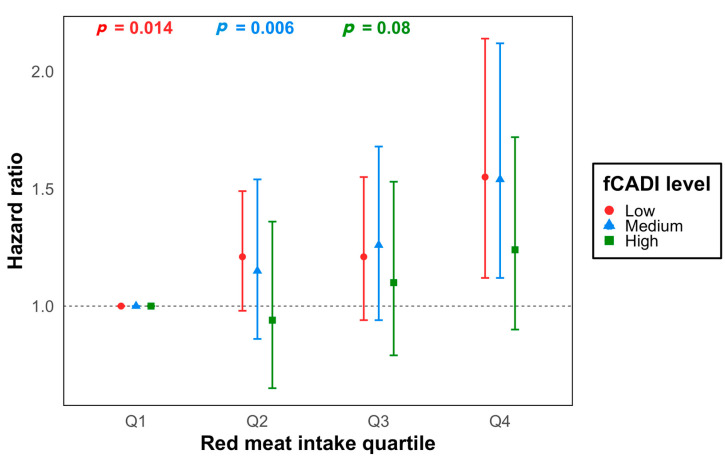
Stratified hazard ratios for lung cancer by red meat intake quartiles and fCADI levels. The model was adjusted for age, sex, race, study arm, body mass index category, education level, marital status, family history of any cancer, total energy intake, pack-years of smoking, alcohol drinks per day, and supplement use.

**Figure 2 antioxidants-13-00799-f002:**
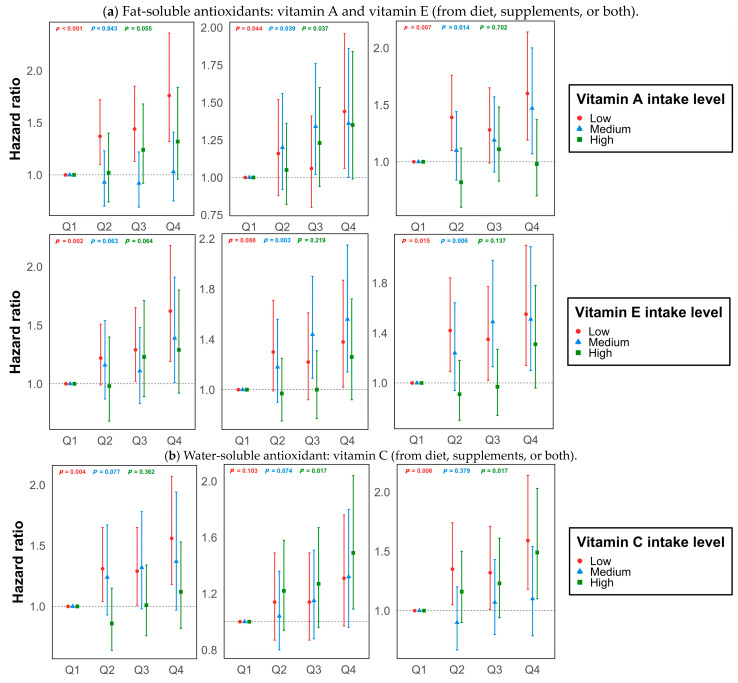
Stratified hazard ratios for lung cancer by red meat quartiles and antioxidant levels. Models were adjusted for age, sex, race, study arm, body mass index category, education level, marital status, family history of any cancer, total energy intake, pack-years of smoking, alcohol drinks per day, fCDAI (for supplemental antioxidants), and supplement use (for dietary and total antioxidant intake). The X-axis represents quartiles of red meat intake. The left column of plots represents hazard ratios stratified by antioxidants from diet; the middle column of plots represents hazard ratios stratified by antioxidants from supplements; and the right column of plots represents hazard ratios stratified by total antioxidant intake.

**Figure 3 antioxidants-13-00799-f003:**
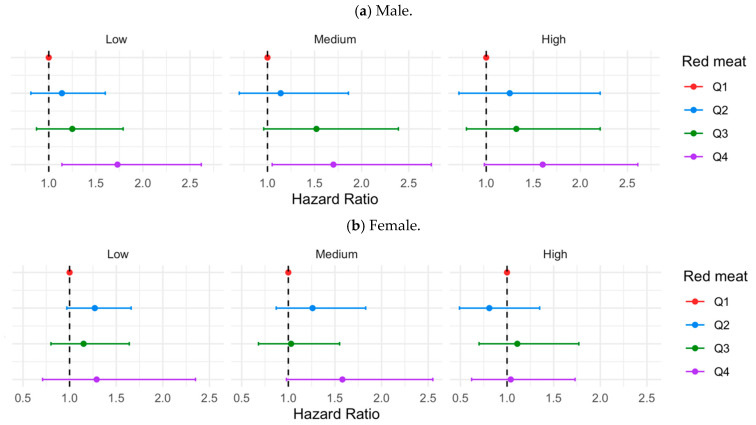
HRs of lung cancer to red meat intake by quartile in subgroups, stratified by three levels of fCDAI. Models were adjusted for age, sex, race, study arm, body mass index category, education level, marital status, family history of any cancer, total energy intake, smoking status, alcohol drinks per day, and supplement use, except for the stratification factors. Labels above each facet represent low, medium, and high levels of fCDAI.

**Table 1 antioxidants-13-00799-t001:** Baseline characteristics of selected participants across quartiles of red meat intake ^1^.

	**Overall** **(N = 95,647)**	**Quartile 1 ** **(N = 23,912)**	**Quartile 2 ** **(N = 23,912)**	**Quartile 3** **(N = 23,912)**	**Quartile 4 ** **(N = 23,911)**
Age, yr	62.3 (5.3)	63.1 (5.5)	62.6 (5.3)	62.3 (5.2)	61.4 (5.0)
Female, n (%)	49,066 (51.3)	17,161 (71.8)	14,510 (60.7)	11,191 (46.8)	6204 (25.9)
Intervention arm, n (%)	48,998 (51.2)	12,029 (50.3)	12,207 (51.0)	12,384 (51.8)	12,378 (51.8)
Race, n (%)					
White, non-Hispanic	87,255 (91.2)	20,664 (86.4)	21,933 (91.7)	22,243 (93.0)	22,415 (93.7)
Black, non-Hispanic	3042 (3.2)	1233 (5.2)	717 (3.0)	576 (2.4)	516 (2.2)
Hispanic	1396 (1.5)	352 (1.5)	303 (1.3)	353 (1.5)	388 (1.6)
Asian	3327 (3.5)	1505 (6.3)	812 (3.4)	589 (2.5)	421 (1.8)
Pacific Islander	439 (0.5)	118 (0.5)	106 (0.4)	107 (0.4)	108 (0.5)
American Indian	188 (0.2)	40 (0.2)	41 (0.2)	44 (0.2)	63 (0.3)
Body mass index category, n (%)					
Underweight	643 (0.7)	286 (1.2)	153 (0.6)	119 (0.5)	85 (0.4)
Normal	32,277 (33.7)	11,024 (46.1)	8802 (36.8)	7125 (29.8)	5326 (22.3)
Overweight	40,840 (42.7)	8702 (36.4)	10,005 (41.8)	10,889 (45.5)	11,244 (47.0)
Obese	21,887 (22.9)	3900 (16.3)	4952 (20.7)	5779 (24.2)	7256 (30.3)
Education level, n (%)					
Less than high school	5554 (5.8)	1149 (4.8)	1282 (5.4)	1382 (5.8)	1741 (7.3)
High school or post-high school training	34,515 (36.1)	7928 (33.2)	8684 (36.3)	8788 (36.8)	9115 (38.1)
Some college or more	55,578 (58.1)	14,835 (62.0)	13,946 (58.3)	13,742 (57.5)	13,055 (54.6)
Marital status, n (%)					
Married	75,124 (78.5)	16,649 (69.6)	18,818 (78.7)	19,608 (82.0)	20,049 (83.8)
Widowed, divorced, or separated	17,501 (18.3)	6253 (26.2)	4401 (18.4)	3653 (15.3)	3194 (13.4)
Never married	3022 (3.2)	1010 (4.2)	693 (2.9)	651 (2.7)	668 (2.8)
Smoke, pack-years	18.0 (26.7)	12.9 (22.1)	15.6 (24.2)	18.9 (27.1)	24.5 (31.3)
Alcohol, drinks/day	0.7 (1.9)	0.4 (1.3)	0.6 (1.6)	0.8 (2.1)	1.1 (2.4)
Family history of any cancer, n (%)	53,402 (55.8)	13,499 (56.5)	13,631 (57.0)	13,232 (55.3)	13,040 (54.5)
Total energy intake, kcal/day	1742 (735)	1295 (517)	1487 (507)	1783 (567)	2405 (782)
fCADI score	0.0 (4.8)	−1.8 (4.4)	−1.4 (3.9)	0.0 (4.0)	3.2 (4.9)
Current supplement use, n (%)	74,846 (78.3)	20,157 (84.3)	19,253 (80.5)	18,360 (76.8)	17,076 (71.4)

^1^ Values of each entry are mean (standard deviation) for continuous variables or n (%) for categorical variables.

**Table 2 antioxidants-13-00799-t002:** HRs of lung cancer to red meat intake by quartile.

	Categories of Red Meat Consumption by Quartile (g/day)				*p*-Trend
	1 (<27)	2 (27–48)	3 (48–81)	4 (>81)	
**Cases**	317	373	406	503	
**Person-Years**	287,600	287,429	285,212	281,951	
Model 1 ^1^	1 (ref)	1.18 (1.01, 1.37)	1.28 (1.10, 1.49)	1.63 (1.41, 1.89)	<0.001
Model 2 ^2^	1 (ref)	1.16 (1.00, 1.35)	1.23 (1.06, 1.44)	1.43 (1.20, 1.71)	<0.001

^1^ Model 1 adjusted for age and sex. ^2^ Model 2 further adjusted for race, study arm, body mass index category, education level, marital status, family history of any cancer, total energy intake, pack-years of smoking, and alcohol drinks per day.

**Table 3 antioxidants-13-00799-t003:** Main and interaction effects of red meat intake and fCDAI to lung cancer risk ^1^.

Main Effect		Interaction Effect			
Red Meat	Hazard Ratio (95% CI)	Red Meat × fCDAI		Hazard Ratio (95% CI)	*p*-Interaction ^2,3^
Q1	1.00 (ref)	Q1	Low	1.00 (ref)	0.773
Q2	1.24 (1.01, 1.52)	Q2	Medium	0.91 (0.64, 1.29)	
			High	0.76 (0.50, 1.15)	
Q3	1.29 (1.03, 1.62)	Q3	Medium	0.94 (0.66, 1.33)	
			High	0.85 (0.57, 1.25)	
Q4	1.73 (1.29, 2.31)	Q4	Medium	0.81 (0.55, 1.19)	
			High	0.70 (0.47, 1.05)	

^1^ Model was adjusted for age, sex, race, study arm, body mass index category, education level, marital status, family history of any cancer, total energy intake, pack-years of smoking, alcohol drinks per day, supplement use, and a product term (quartiles of red meat intake and levels of antioxidant intake). ^2^ Likelihood ratio test was used to examine the statistical interaction between red meat and fCDAI intake. ^3^ Bonferroni-adjusted significance threshold = 0.05/3 (3 levels of antioxidant) = 0.017.

**Table 4 antioxidants-13-00799-t004:** Main and interaction effects of red meat and different sources of antioxidant intake to lung cancer risk ^1,2,3^.

(**1a**) Dietary vitamin A.					
**Main effect**		**Interaction effect**			
Red meat	Hazard ratio (95% CI)	Red meat × dietary vitamin A		Hazard ratio (95% CI)	*p*-interaction
Q1	1.00 (ref)	Q1	Low	1.00 (ref)	0.494
Q2	1.33 (1.06, 1.66)	Q2	Medium	0.73 (0.51, 1.04)	
			High	0.78 (0.53, 1.14)	
Q3	1.34 (1.06, 1.68)	Q3	Medium	0.74 (0.53, 1.05)	
			High	0.95 (0.66, 1.37)	
Q4	1.57 (1.22, 2.01)	Q4	Medium	0.75 (0.53, 1.05)	
			High	0.86 (0.61, 1.22)	
(**1b**) Supplemental vitamin A.					
**Main effect**		**Interaction effect**			
Red meat	Hazard ratio (95% CI)	Red meat × supplemental vitamin A		Hazard ratio (95% CI)	*p*-interaction
Q1	1.00 (ref)	Q1	Low	1.00 (ref)	0.636
Q2	1.17 (0.89, 1.53)	Q2	Medium	1.05 (0.73, 1.53)	
			High	0.91 (0.63, 1.32)	
Q3	1.09 (0.83, 1.43)	Q3	Medium	1.31 (0.90, 1.89)	
			High	1.18 (0.82, 1.69)	
Q4	1.46 (1.11, 1.91)	Q4	Medium	1.01 (0.71, 1.43)	
			High	1.00 (0.71, 1.42)	
(**1c**) Total vitamin A.					
**Main effect**		**Interaction effect**			
Red meat	Hazard ratio (95% CI)	Red meat × total vitamin A		Hazard ratio (95% CI)	*p*-interaction
Q1	1.00 (ref)	Q1	Low	1.00 (ref)	0.068
Q2	1.35 (1.07, 1.71)	Q2	Medium	0.84 (0.59, 1.19)	
			High	0.62 (0.42, 0.91)	
Q3	1.20 (0.94, 1.53)	Q3	Medium	1.05 (0.74, 1.49)	
			High	0.96 (0.67, 1.39)	
Q4	1.43 (1.12, 1.84)	Q4	Medium	1.10 (0.79, 1.54)	
			High	0.74 (0.53, 1.05)	
(**2a**) Dietary vitamin C.					
**Main effect**		**Interaction effect**			
Red meat	Hazard ratio (95% CI)	Red meat × dietary vitamin C		Hazard ratio (95% CI)	*p*-interaction
Q1	1.00 (ref)	Q1	Low	1.00 (ref)	0.394
Q2	1.29 (1.03, 1.62)	Q2	Medium	0.98 (0.68, 1.41)	
			High	0.67 (0.46, 0.96)	
Q3	1.26 (0.99, 1.59)	Q3	Medium	1.09 (0.76, 1.56)	
			High	0.81 (0.57, 1.15)	
Q4	1.51 (1.18, 1.92)	Q4	Medium	0.97 (0.68, 1.37)	
			High	0.76 (0.55, 1.06)	
(**2b**) Supplemental vitamin C.					
**Main effect**		**Interaction effect**			
Red meat	Hazard ratio (95% CI)	Red meat × supplemental vitamin C		Hazard ratio (95% CI)	*p*-interaction
Q1	1.00 (ref)	Q1	Low	1.00 (ref)	0.854
Q2	1.15 (0.88, 1.49)	Q2	Medium	0.93 (0.64, 1.34)	
			High	1.09 (0.75, 1.57)	
Q3	1.17 (0.90, 1.52)	Q3	Medium	1.05 (0.73, 1.51)	
			High	1.17 (0.82, 1.69)	
Q4	1.32 (1.01, 1.72)	Q4	Medium	1.13 (0.80, 1.59)	
			High	1.23 (0.87, 1.75)	
(**2c**) Total vitamin C.					
**Main effect**		**Interaction effect**			
Red meat	Hazard ratio (95% CI)	Red meat × total vitamin C		Hazard ratio (95% CI)	*p*-interaction
Q1	1.00 (ref)	Q1	Low	1.00 (ref)	0.575
Q2	1.34 (1.04, 1.71)	Q2	Medium	0.68 (0.47, 1.00)	
			High	0.87 (0.61, 1.24)	
Q3	1.27 (0.99, 1.63)	Q3	Medium	0.89 (0.62, 1.29)	
			High	0.97 (0.68, 1.39)	
Q4	1.45 (1.12, 1.88)	Q4	Medium	0.87 (0.61, 1.23)	
			High	1.02 (0.73, 1.43)	
(**3a**) Dietary vitamin E.					
**Main effect**		**Interaction effect**			
Red meat	Hazard ratio (95% CI)	Red meat × dietary vitamin E		Hazard ratio (95% CI)	*p*-interaction
Q1	1.00 (ref)	Q1	Low	1.00 (ref)	0.770
Q2	1.22 (0.99, 1.50)	Q2	Medium	0.97 (0.68, 1.37)	
			High	0.79 (0.53, 1.20)	
Q3	1.28 (1.02, 1.60)	Q3	Medium	0.89 (0.63, 1.26)	
			High	0.94 (0.64, 1.37)	
Q4	1.58 (1.21, 2.07)	Q4	Medium	0.89 (0.61, 1.28)	
			High	0.78 (0.53, 1.15)	
(**3b**) Supplemental vitamin E.					
**Main effect**		**Interaction effect**			
Red meat	Hazard ratio (95% CI)	Red meat × supplemental vitamin E		Hazard ratio (95% CI)	*p*-interaction
Q1	1.00 (ref)	Q1	Low	1.00 (ref)	0.247
Q2	1.32 (1.01, 1.73)	Q2	Medium	0.92 (0.63, 1.35)	
			High	0.74 (0.51, 1.07)	
Q3	1.28 (0.98, 1.68)	Q3	Medium	1.19 (0.82, 1.73)	
			High	0.80 (0.56, 1.16)	
Q4	1.45 (1.10, 1.90)	Q4	Medium	1.19 (0.84, 1.70)	
			High	0.90 (0.64, 1.28)	
(**3c**) Total vitamin E.					
**Main effect**		**Interaction effect**			
Red meat	Hazard ratio (95% CI)	Red meat × total vitamin E		Hazard ratio (95% CI)	*p*-interaction
Q1	1.00 (ref)	Q1	Low	1.00 (ref)	0.045
Q2	1.41 (1.08, 1.82)	Q2	Medium	0.90 (0.62, 1.32)	
			High	0.63 (0.44, 0.91)	
Q3	1.32 (1.01, 1.71)	Q3	Medium	1.19 (0.82, 1.72)	
			High	0.70 (0.49, 1.01)	
Q4	1.47 (1.12, 1.92)	Q4	Medium	1.15 (0.81, 1.65)	
			High	0.83 (0.59, 1.17)	
(**4a**) Dietary zinc.					
**Main effect**		**Interaction effect**			
Red meat	Hazard ratio (95% CI)	Red meat × dietary zinc		Hazard ratio (95% CI)	*p*-interaction
Q1	1.00 (ref)	Q1	Low	1.00 (ref)	0.097
Q2	1.17 (0.96, 1.42)	Q2	Medium	1.02 (0.71, 1.46)	
			High	0.91 (0.58, 1.43)	
Q3	1.34 (1.06, 1.70)	Q3	Medium	0.99 (0.69, 1.42)	
			High	0.77 (0.49, 1.17)	
Q4	2.18 (1.36, 3.51)	Q4	Medium	0.85 (0.49, 1.48)	
			High	0.50 (0.28, 0.90)	
(**4b**) Supplemental zinc.					
**Main effect**		**Interaction effect**			
Red meat	Hazard ratio (95% CI)	Red meat × supplemental zinc		Hazard ratio (95% CI)	*p*-interaction
Q1	1.00 (ref)	Q1	Low	1.00 (ref)	0.460
Q2	1.02 (0.79, 1.33)	Q2	Medium	1.19 (0.82, 1.71)	
			High	1.20 (0.83, 1.74)	
Q3	1.04 (0.80, 1.34)	Q3	Medium	1.30 (0.91, 1.86)	
			High	1.38 (0.96, 1.97)	
Q4	1.25 (0.96, 1.62)	Q4	Medium	1.15 (0.82, 1.62)	
			High	1.42 (1.01, 1.99)	
(**4c**) Total zinc.					
**Main effect**		**Interaction effect**			
Red meat	Hazard ratio (95% CI)	Red meat × total zinc		Hazard ratio (95% CI)	*p*-interaction
Q1	1.00 (ref)	Q1	Low	1.00 (ref)	0.311
Q2	1.22 (0.97, 1.53)	Q2	Medium	0.82 (0.58, 1.16)	
			High	1.08 (0.73, 1.60)	
Q3	1.11 (0.87, 1.41)	Q3	Medium	1.18 (0.83, 1.66)	
			High	1.27 (0.87, 1.85)	
Q4	1.46 (1.12, 1.91)	Q4	Medium	0.86 (0.60, 1.22)	
			High	1.14 (0.78, 1.66)	
(**5a**) Dietary magnesium.					
**Main effect**		**Interaction effect**			
Red meat	Hazard ratio (95% CI)	Red meat × dietary magnesium		Hazard ratio (95% CI)	*p*-interaction
Q1	1.00 (ref)	Q1	Low	1.00 (ref)	0.276
Q2	1.22 (0.98, 1.52)	Q2	Medium	1.04 (0.74, 1.46)	
			High	0.75 (0.50, 1.12)	
Q3	1.39 (1.04, 1.78)	Q3	Medium	0.80 (0.56, 1.13)	
			High	0.85 (0.59, 1.24)	
Q4	1.31 (0.94, 1.81)	Q4	Medium	1.15 (0.76, 1.72)	
			High	1.01 (0.67, 1.53)	
(**5b**) Supplemental magnesium.					
**Main effect**		**Interaction effect**			
Red meat	Hazard ratio (95% CI)	Red meat × supplemental magnesium		Hazard ratio (95% CI)	*p*-interaction
Q1	1.00 (ref)	Q1	Low	1.00 (ref)	0.339
Q2	1.08 (0.83, 1.40)	Q2	Medium	1.23 (0.85, 1.77)	
			High	1.00 (0.69, 1.44)	
Q3	1.11 (0.86, 1.44)	Q3	Medium	1.12 (0.78, 1.61)	
			High	1.31 (0.92, 1.86)	
Q4	1.28 (0.98, 1.67)	Q4	Medium	1.20 (0.85, 1.70)	
			High	1.27 (0.90, 1.80)	
(**5c**) Total magnesium.					
**Main effect**		**Interaction effect**			
Red meat	Hazard ratio (95% CI)	Red meat × total magnesium		Hazard ratio (95% CI)	*p*-interaction
Q1	1.00 (ref)	Q1	Low	1.00 (ref)	0.835
Q2	1.31 (1.04, 1.64)	Q2	Medium	0.80 (0.56, 1.13)	
			High	0.84 (0.57, 1.23)	
Q3	1.27 (0.99, 1.62)	Q3	Medium	0.93 (0.66, 1.32)	
			High	0.98 (0.67, 1.42)	
Q4	1.42 (1.05, 1.91)	Q4	Medium	1.02 (0.70, 1.49)	
			High	0.98 (0.67, 1.45)	
(**6a**) Dietary selenium.					
**Main effect**		**Interaction effect**			
Red meat	Hazard ratio (95% CI)	Red meat × dietary selenium		Hazard ratio (95% CI)	*p*-interaction
Q1	1.00 (ref)	Q1	Low	1.00 (ref)	0.117
Q2	1.29 (1.06, 1.57)	Q2	Medium	0.75 (0.53, 1.05)	
			High	0.83 (0.51, 1.35)	
Q3	1.51 (1.19, 1.93)	Q3	Medium	0.65 (0.46, 0.93)	
			High	0.81 (0.52, 1.28)	
Q4	1.43 (0.82, 2.52)	Q4	Medium	1.02 (0.55, 1.91)	
			High	0.90 (0.46, 1.75)	
(**6b**) Supplemental selenium.					
**Main effect**		**Interaction effect**			
Red meat	Hazard ratio (95% CI)	Red meat × supplemental selenium		Hazard ratio (95% CI)	*p*-interaction
Q1	1.00 (ref)	Q1	Low	1.00 (ref)	0.913
Q2	1.11 (0.85, 1.44)	Q2	Medium	1.04 (0.72, 1.51)	
			High	1.11 (0.77, 1.60)	
Q3	1.24 (0.95, 1.62)	Q3	Medium	1.06 (0.74, 1.52)	
			High	0.99 (0.69, 1.42)	
Q4	1.48 (1.13, 1.94)	Q4	Medium	0.93 (0.65, 1.31)	
			High	1.06 (0.75, 1.50)	
(**6c**) Total selenium.					
**Main effect**		**Interaction effect**			
Red meat	Hazard ratio (95% CI)	Red meat × total selenium		Hazard ratio (95% CI)	*p*-interaction
Q1	1.00 (ref)	Q1	Low	1.00 (ref)	0.102
Q2	1.24 (1.02, 1.51)	Q2	Medium	0.85 (0.60, 1.19)	
			High	0.87 (0.54, 1.39)	
Q3	1.51 (1.20, 1.91)	Q3	Medium	0.67 (0.47, 0.96)	
			High	0.87 (0.56, 1.35)	
Q4	1.29 (0.75, 2.23)	Q4	Medium	1.21 (0.66, 2.22)	
			High	1.05 (0.55, 2.01)	

^1^ Models were adjusted for age, sex, race, study arm, body mass index category, education level, marital status, family history of any cancer, total energy intake, pack-years of smoking, alcohol drinks per day, fCDAI (for supplemental antioxidants), supplement use (for dietary and total antioxidant intake), and a product term (quartiles of red meat intake and levels of antioxidant intake). ^2^ Likelihood ratio test was used to examine the statistical interaction between red meat and antioxidant intake. ^3^ Bonferroni-adjusted significance threshold = 0.05/3 (3 levels of antioxidant) = 0.017.

## Data Availability

Data described in the manuscript, code book, and analytic code will not be made available because of participant confidentiality and privacy concerns. The PLCO lung dataset used in this study is available based on the request to the NCI Cancer Data Access System.

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
