# Peer review of "Modification Role of Dietary Antioxidants in the Association of High Red Meat Intake and Lung Cancer Risk: Evidence from a Cancer Screening Trial"

_antioxidants, 2024, doi:10.3390/antiox13070799_

Round 1

Reviewer 1 Report

According to current status of knowledge high red meat consumption may be combined with the increased lung cancer risk. The data presented by the authors confirm this rule. Nevertheless, several questions may be asked: is the effect of high red meat intake independent on age, weight, sex and other variables? Is the definition of high intake the same for the individuals under and overweighted? Which of the dietary compounds should be recommended due to their antioxidative potency? Is it better to use natural or syntetic antioxidants, or both? In this respect the presented data are inconclusive, despite advanced statistical methods applied

The original paper entitled „Modification role of dietary antioxidants in the association of high red meat intake and lung cancer risk: evidence from a cancer screening trial” concerns the influence of red meat consumption of lung cancer risk, and the possibility of its modification with  natural and/or  synthetic antioxidants consumption.

The  advantage of the study is the large group of participants (95647 persons included in PLCO screening trial) and prolonged observation time (13 years). 1599 lung cancer cases developed during the follow up.

High red meat consumers (>81g/day) were more likely men and overweight/obese persons. The authors proved that the higher red meat consumption was associated with the increased lung cancer risk,  also if the data were adjusted for age and sex, but not for personal weight.

The data concerning the modification of such risk with dietary and/or synthetic antioxidants consumption were  less  consistent. In my opinion, it would be better to analyze all the factors influencing lung cancer risk in multivariate analysis,  using Cox proportional hazard models.  The dietary intake of  six major antioxidants was standardized (fCDAI), therefore it is reasonable to use this index and not to speculate on the role of single antioxidants, especially that they haven’t been used as single agents.  Instead of dividing the antioxidant consumption into low, medium and high,  I would advise to use the amount of  antioxidants consumed per day.

It is interesting,  that the reduced  risk of lung cancer in persons declaring high red meat intake, if they used high antioxidant prophylaxis,  was observed in females but not in males. It may depend on small number of women involved in this analysis, as only 26% of females declared high red-meat consumption.

The conclusions are not fully supported  by the results of the study,  as the protective effect of antioxidants was probably dependent on the type of antioxidant and on population characteristics.  

I think that the study should be published but major revision is needed to obtain more precise results.

Reviewer 2 Report

The subject is of some clinical interest and the paper is in general well written, but some points should be detailed/clarified and the ideas more organized and systematic.

1. Why did the authors choose lung cancer as model to evaluate the effect of red meat and antioxidants’ role in its progression? As lungs are not involved in digestion and absorption of red meat, it was expected that red meat could be a risk factor for this type of cancer? How can you explain that?

2. In lane 66, it is referred that “To the best of our knowledge, no study has studied the potential modification effects of anti-oxidants from different sources on associations between red meat and lung cancer risk”. The authors should clarify here in a better way what is the objective of the work, as it is not very clear.

3. Why can the increased consumption of red meat lead to a higher risk of cancer? What is the mechanism associated? What is the role of the antioxidants. This should be further detailed in the beginning of the introduction

4, Several antioxidants were assayed in this work. What is the difference between their mechanism of action and how can that be related to the results?

5. Can the authors explain how can tobacco and red meat be associated?

6. It is difficult to know what the main message of the paper and what novelty it brings. The authors should emphasize such message and evidence the novelty.

no further comments

Round 2

Reviewer 1 Report

The Authors answered my remarks. I have no further questions.

The Authors answered my remarks. I have no further questions.

Reviewer 2 Report

The authors answered to the questions made and the paper is now in conditions to be published.

The authors answered to the questions made and the paper is now in conditions to be published.